# Effects on Cell Membrane Integrity of *Pichia anomala* by the Accumulating Excessive Reactive Oxygen Species under Ethanol Stress

**DOI:** 10.3390/foods11223744

**Published:** 2022-11-21

**Authors:** Yanru Chen, Yin Wan, Wenqin Cai, Na Liu, Jiali Zeng, Chengmei Liu, Hong Peng, Guiming Fu

**Affiliations:** 1State Key Laboratory of Food Science and Technology, College of Food Science and Technology, Nanchang University, Nanchang 330047, China; 2International Institute of Food Innovation, Nanchang University, Nanchang 330299, China

**Keywords:** ethanol stress, *Pichia anomala*, ester production ability, reactive oxygen species, cell membrane integrity

## Abstract

Ethanol stress to yeast is well recognized and exists widely during the brewing process of alcohol products. *Pichia anomala* is an important ester-producing yeast in the brewing process of Chinese *Baijiu* and other alcohol products. Therefore, it is of great significance for the alcohol products brewing industry to explore the effects of ethanol stress on the growth metabolism of *P. anomala*. In this study, the effects of ethanol stress on the growth, esters production ability, cell membrane integrity and reactive oxygen species (ROS) metabolism of *P. anomala* NCU003 were studied. Our results showed that ethanol stress could inhibit the growth, reduce the ability of non-ethyl ester compounds production and destroy the cell morphology of *P. anomala* NCU003. The results also showed that 9% ethanol stress produced excessive ROS and then increased the activities of antioxidant enzymes (superoxide dismutase, catalase, aseorbateperoxidase and glutathione reductase) compared to the control group. However, these increased antioxidant enzyme activities could not prevent the damage caused by ROS to *P. anomala* NCU003. Of note, correlation results indicated that high content of ROS could promote the accumulation of malondialdehyde content, resulting in destruction of the integrity of the cell membrane and leading to the leakage of intracellular nutrients (soluble sugar and protein) and electrolytes. These results indicated that the growth and the non-ethyl ester compounds production ability of *P. anomala* could be inhibited under ethanol stress by accumulating excessive ROS and the destruction of cell membrane integrity in *P. anomala*.

## 1. Introduction

Yeast is one of the three major microbial communities (yeast, bacteria and mold) and plays an important role in the generation of ethanol and flavor compounds during the brewing process of various alcohol products [1,2,3]. Among them, *Saccharomyces cerevisiae* metabolizes sugars in fruits, grains and other raw materials into ethanol; ester-producing yeast generates many flavor compounds, including acids, aldehydes and esters, giving alcohol products different flavor profiles [4]. Of note, *Pichia anomala* is an important ester-producing yeast during the brewing process of alcohol products [5,6]. Studies have indicated that *P. anomala* has an important contribution to improving the content of ethyl acetate, ethyl hexanoate and 2-phenylethyl acetate in Chinese *Baijiu* and wine [7,8,9]. In our previous study, we also found that *P. anomala* was the predominant ester-producing yeast during the brewing process of Chinese special-flavor *Baijiu* [10,11].

However, the brewing process of alcohol products has always been a complex system with dynamic environmental changes of physicochemical factors. With the increase in ethanol concentration during the brewing process, yeast is susceptible to ethanol stress and it reduces the fermentation efficiency, leading to the decline of quality and flavor profiles in the finished alcohol products [12,13]. For example, the ethanol stress could inhibit the growth of *S. cerevisiae* during the wine brewing process, resulting in prolonging the brewing cycle and affecting the production of flavor compounds in wine [14,15]. Lu et al. also found that the physiological metabolic activities of *S. cerevisiae* were gradually inhibited by ethanol stress during the grain fermentation process of Chinese *Baijiu*, and then the ability to convert and utilize raw materials was decreased markedly [16]. Meanwhile, studies have found that ester-producing yeast is more susceptible to ethanol stress compared with *S. cerevisiae* [17]. Therefore, it has become a hotspot to study the effects of ethanol stress on the growth and esters production ability of ester-producing yeast.

Reactive oxygen species (ROS) is a by-product that is generated in yeast during the process of aerobic metabolism. Studies have indicated that high content of ROS could destroy the DNA structure, affect the synthesis of proteins and cause cell membrane oxidation, thereby inhibiting the growth of *S. cerevisiae* [4,18]. In addition, Mahmud et al. found that environmental stress could cause the evacuation of cell skeleton, destroy cell membrane integrity and lead to leakage of intracellular electrolytes and nutrients, thereby reducing the growth and fermentation characteristics of *S. cerevisiae* [19]. Interestingly, studies have also found that the change in intracellular ROS content is related to the change in cell membrane integrity when the *S. cerevisiae* was faced with environmental factor stress [20,21]. However, the growth effect and inhibition mechanism of *P. anomala* under ethanol stress still remains unknown.

Our previous research found that there was a negative correlation between the changes of *Pichia* genus and ethanol concentration in fermented grain during the brewing process of Chinese special-flavor *Baijiu*, and the highest concentration of ethanol could reach 9% [10]. Therefore, to explore the effect of ethanol stress on the growth, ester production ability and influence mechanism of *P. anomala*, one goal of this study is to describe the effects of ethanol stress on the growth and esters production ability of *P. anomala*; the other goal is to clarify the potential inhibition mechanism of ethanol stress and the effect on ROS metabolism and cell membrane integrity of *P. anomala*. It is hoped that this will provide a better theoretical basis for the application of *P. anomala* in the alcohol products brewing industry.

## 2. Materials and Methods

### 2.1. Strain

The *P. anomala* NCU003 strain with high yield esters capability was originally isolated according to a culture-dependent method from special-flavor *Baijiu Daqu* [11]. It was maintained on yeast extract peptone dextrose (YPD) medium (1% yeast extract, 2% peptone, 2% glucose) (Huankai Microbial Technology Co., Ltd., Guangzhou, China) at 28 °C.

### 2.2. Culture Conditions

*P. anomala* NCU003 was pre-cultured overnight in 100 mL YPD medium (28 °C, 160 r/min); one (1.0) mL pre-cultured *P. anomala* NCU003 medium (the cell concentration was adjusted to 0.1 of OD_600_) was transferred to a new 100 mL YPD medium (determination of growth-related indicators) or malt extract medium (determination of flavor compounds) (Huankai Microbial Technology Co., Ltd., Guangzhou, China), then cultured for 8 h (28 °C, 160 r/min). Finally, ethanol was added into medium to make YPD ethanol (YPDE) medium and malt extract ethanol (MEE) medium, respectively. The initial concentration of ethanol was 3%, 6% and 9% (*v/v*), and no additional ethanol was added into the control group. In addition, all experiments with parallel samples were repeated three times and were determined every 2 h (0, 2, 4, 6, 8, 10, 12, 14 h) after adding ethanol stress.

### 2.3. Determination of Growth of P. anomala NCU003 under Ethanol Stress

Ten (10) mL YPDE medium was taken and the OD_600_ value was measured. The time as the horizontal coordinate, the OD_600_ value as a longitudinal coordinate and the growth curve of *P. anomala* NCU003 was drawn. The spot assay method referred to the method of Mitsui et al. [22]. Five (5.0) mL YPD medium after 8 h of preculture (the cell concentration was adjusted to 0.1 of OD_600_) was diluted to 10^−5^, 10^−4^ and 10^−3^ cell suspension, then 2.0 μL suspension was added to YPD agar medium, which contained ethanol at different concentrations (3%, 6% and 9%), and incubated at 28 °C for 36 h. Finally, the effect of colony growth of *P. anomala* NCU003 in solid medium with different ethanol stress concentrations was observed.

### 2.4. Determination of Esters Production Ability of P. anomala NCU003 by HS-SPME-GC-MS

Eight (8.0) mL of MEE medium after centrifugation (4 °C, 12,000× *g*, 10 min) was poured into 15 mL sample bottle, followed by addition of 0.8 g NaCl and 80 μL 4-methyl-2-pentanol internal standard solution (400 μg/L). The subsequent operation was according to the method of Fu et al. [23]. GC (Agilent Technologies, Santa Clara, CA, USA) conditions: HP-5MS column (30 m × 0.25 mm × 0.25 μm); carrier gas: He; flow rate: 1 mL/min; solvent delay: 0.5 min; split ratio: 15:1; temperature program: 35 °C/5 min–(8 °C/min)–96 °C/3 min–(2 °C/min)–110 °C–(15 °C/min)–240 °C/3 min. MS conditions: electron energy: 70 eV; ion source temperature: 230 °C; inlet temperature: 250 °C; scanning range: 33–450 *m/z*; scanning mode: full scan. The methods of qualitative and quantitative were according to the method of Ulrich et al. [24].

### 2.5. Determination of Cell Surface Morphology of P. anomala NCU003 with Scanning Electron Microscopy (SEM)

The pretreatment of SEM was according to the method of Miao et al. with a slight modification [25]. Five (5.0) mL YPDE medium after 4 h of stress was taken and *P. anomala* NCU003 cells were collected after centrifugation (4 °C, 12,000× *g*, 10 min). Cells were resuspended in 2% glutaraldehyde overnight and then washed 3 times with phosphate buffered saline (PBS) solution for 10 min per wash; the above steps were carried out at 4 °C. The next steps were dehydration, replacement, vacuum drying and gold-plating, respectively. Finally, the surface morphology of *P. anomala* NCU003 was observed and pictures were taken with JSM-6701F scanning electron microscopy system (JEOL Ltd., Akishima City, Japan).

### 2.6. Determination of Cell Membrane Integrity of P. anomala NCU003 with Propidium Iodide (PI) Staining

The pretreatment of PI staining was according to the method of Zhang et al. with a slight modification [26]. Five (5.0) mL YPDE medium was taken after 4 h of stress and *P. anomala* NCU003 cells were collected after centrifugation (4 °C, 12,000× *g*, 10 min). Cells were resuspended in PBS solution after washing 3 times with PBS solution. The subsequent operation was according to the manufacturer’s instructions for the PI kit (Solarbio Science and Technology Co., Ltd., Beijing, China). Finally, the PI staining results of *P. anomala* NCU003 were observed and pictures were taken with Leica inverted fluorescent microscope (Leica Microsystems, Wetzlar, Germany).

### 2.7. Determination of Intracellular Malondialdehyde (MDA) and Extracellular Leakage Substance Content of P. anomala NCU003

The content of intracellular MDA was measured according to the method of Wei et al. with a slight modification [27]. Ten (10) mL YPDE medium was taken and *P. anomala* NCU003 cells were collected after centrifugation (4 °C, 12,000× *g*, 10 min). Five (5.0) mL trichloroacetic acid solution was added and the cell solution was broken for 10 min by ultrasound. The supernatant was then collected after centrifugation (4 °C, 12,000× *g*, 10 min), and then 2.0 mL supernatant was mixed in 2.0 mL thiolabitturic acid solution and boiled for 15 min. Absorbance of the mixture after cooling was measured at 450 nm, 532 nm and 600 nm, respectively, and the content of MDA was expressed as μmol/g.

The extracellular electrical conductivity was measured according to the method of Liu et al., and the result of electrical conductivity was expressed as ms/cm [28]. The content of extracellular soluble protein was measured according to the method of Bradford, and the content of soluble protein was expressed as μg/g [29]. The content of extracellular soluble sugar was measured according to the method of Morris, and the content of soluble sugar was expressed as mg/g [30].

### 2.8. Determination of ROS Content of P. anomala NCU003

Five (5.0) mL YPDE medium was taken and *P. anomala* NCU003 cells were washed 3 times with PBS solution and were collected after centrifugation (4 °C, 12,000× *g*, 10 min). The subsequent operation was according to the manufacturer’s instructions for the Active oxygen detection kit (Shanghai Beyotime Biotechnology Co., Ltd., Shanghai, China). The relative fluorescence intensity of *P. anomala* NCU003 cells was measured by F-7000 fluorescence spectrophotometer (HITACHI Co., Ltd., Chiyoda City, Japan) at Ex = 488 nm and Em = 525 nm. The content of ROS was expressed as relative fluorescence intensity, where the higher the relative fluorescence intensity, the higher the ROS content.

### 2.9. Determination of Antioxidant Enzyme Activity of P. anomala NCU003

Ten (10) mL YPDE medium was taken and *P. anomala* NCU003 cells were collected after centrifugation (4 °C, 12,000× *g*, 10 min). Cells were resuspended in PBS solution and washed 3 times and then placed on ice ultrasonic crush for 300 s (60 times, ultrasound 5 s, suspend 5 s). Finally, the supernatant was collected and used for the antioxidant enzymes activity determination.

The activity of aseorbateperoxidase (APX) was measured according to the method of Ge et al. [31]. The activities of superoxide dismutase (SOD), catalase (CAT) and glutathione reductase (GR) were measured by following the manufacturer’s instructions for teh related antioxidant enzymes activity analysis kit (Naning Jiancheng Bioengineering Institute, Nanjing, China). The activities of antioxidant enzymes were all expressed as U/g.

### 2.10. Data Analysis

Microsoft Excel 2019 (Microsoft Corporation, Redmond City, DC, USA) and SPSS 21.0 (SPSS Software Development Co., Ltd., Chicago City, IL, USA) were used to perform all statistical analysis. Single factor analysis and Duncan’s test were used to perform significant difference analysis (*p* < 0.05) by single factor analysis and Duncan’s test. Origin 2021 (OriginLab, Northampton, MA, USA) was used to perform correlation analysis by Pearson method.

## 3. Results

### 3.1. Effect of Ethanol Stress on the Growth of P. anomala NCU003

The result of ethanol stress on the growth of *P. anomala* NCU003 in liquid medium is shown in Figure 1A; the inhibitory effect gradually increased with the increase in ethanol stress concentration. Among them, the 9% ethanol stress group almost completely inhibited the growth of *P. anomala* NCU003. The result of ethanol stress on the growth of *P. anomala* NCU003 in solid medium is also shown in Figure 1B; *P. anomala* NCU003 could grow normally in the control group, the colonies were dense, and the single colony was large and plump. However, the colonies intensity of *P. anomala* NCU003 decreased gradually after ethanol stress, and the single colony was small. Especially, the colony growth of *P. anomala* NCU003 in the 9% ethanol stress group was severely inhibited.

### 3.2. Effect of Ethanol Stress on the Esters Production Ability of P. anomala NCU003

A total of 22 kinds of flavor compounds were detected by GC-MS (Appendix A). Among them, the content changes of nine kinds of ester compounds are shown in Figure 2, presenting two completely different results. A total of five kinds of ethyl ester compounds were produced by *P. anomala* NCU003; these ethyl ester compounds’ content increased in *P. anomala* NCU003 with the increase in ethanol stress concentration (Figure 2A–E). The highest content of ethyl acetate, ethyl propanoate, ethyl hexanoate, ethyl caprylate and ethyl caprate in the 9% ethanol stress group was 2.48, 4.97, 3.90, 27.86 and 9.69 times higher than that of control group, respectively. However, the contents of the other four kinds of ester compounds decreased in *P. anomala* NCU003 with the increase in ethanol stress concentration (Figure 2F–I). The highest content of isoamyl formate, phenylethyl acetate, isopentyl acetate and isobutyl acetate in the control group was 4.18, 2.21, 4.62 and 2.64 times higher than that of the 9% ethanol stress group, respectively.

### 3.3. Effect of Ethanol Stress on the Cell Morphology of P. anomala NCU003

As shown in Figure 3, the SEM images of the control group clearly exhibited that the cell surface morphology of *P. anomala* NCU003 was plump and smooth. However, the surface morphology of *P. anomala* NCU003 began to show folds and depressions under the 3% ethanol stress. With the further increase in ethanol concentration, the depressions and folds in the cell surface morphology of *P. anomala* NCU003 became more serious. Among them, the cell surface morphology of *P. anomala* NCU003 was significantly deformed under the 9% ethanol stress. These results indicated that the cell surface morphology of *P. anomala* NCU003 was severely damaged by ethanol stress, and the damage became more and more serious with the increase in ethanol concentration.

### 3.4. Effect of Ethanol Stress on the Cell Membrane Integrity of P. anomala NCU003

As shown in Figure 4, many *P. anomala* NCU003 cells can be observed in the bright field. However, only a few *P. anomala* NCU003 cells were dyed red when the control group was observed in the dark field, and the amount of red cells in the 3% ethanol stress group was slightly higher than that of the control group. However, more than half of *P. anomala* NCU003 cells were dyed red in both the 6% and 9% ethanol stress groups. The results indicated that the cell membrane integrity of *P. anomala* NCU003 decreased as ethanol stress concentration increased.

### 3.5. Effect of Ethanol Stress on the MDA and Extracellular Leakage Substance Content of P. anomala NCU003

As shown in Table 1, with the increase in ethanol stress concentration, the MDA content and electrical conductivity increased in *P. anomala* NCU003. Among them, the content of MDA in the 3% ethanol stress group was significantly higher (*p* < 0.05) than that of the control group at 2, 4 and 12 h. The electrical conductivity in the 3% ethanol stress group was only significantly higher (*p* < 0.05) than that of the control group at 8 and 14 h. However, compared with the control and the 3% ethanol stress groups, the MDA content and electrical conductivity of *P. anomala* NCU003 in the 6% and 9% ethanol stress groups were significantly higher (*p* < 0.05) than them for almost the entire stress period. Among them, the highest content of MDA in the 9% ethanol stress group was determined at 6 h, which was 5.80 times higher than that of the control group; the highest level of electrical conductivity in the 9% ethanol stress group was determined at 2 h, which was 3.21 times higher than that of the control group.

Similarly, with the increase in ethanol stress concentration, the contents of soluble protein and soluble sugar increased in *P. anomala* NCU003. However, with the stress time increased, the contents of soluble protein and soluble sugar decreased in *P. anomala* NCU003. In addition, the contents of soluble protein in the 3%, 6% and 9% ethanol stress groups were significantly higher (*p* < 0.05) than that of the control group for almost the entire stress period. The content of soluble sugar in the 3% ethanol stress group was only significantly higher (*p* < 0.05) than that of the control group at 2 h. Compared with the control and 3% ethanol stress groups, the contents of soluble sugar in the 6% and 9% ethanol groups were significantly higher (*p* < 0.05) than them for almost the entire stress period. Among them, the highest difference of soluble protein and soluble sugar contents in the 9% ethanol stress group for both were determined at 4 h, which was 3.81 times and 24.98 times higher than that of the control group, respectively.

### 3.6. Effect of Ethanol Stress on the ROS Content of P. anomala NCU003

As shown in Figure 5, with the increase in ethanol stress concentration, the content of ROS increased in *P. anomala* NCU003. The overall change in the ROS content increased first at 6 h and then declined for the remaining stress period, except for the control group where the ROS content maintained lower and stable. The ROS content of *P. anomala* NCU003 in the 6% ethanol stress group was significantly lower (*p* < 0.05) than that of the control and 3% ethanol stress groups at 2 and 4 h, and it was significantly higher than (*p* < 0.05) the control and 3% ethanol groups after 8 h. In addition, compared with the control, 3% and 6% ethanol stress groups, the ROS content of *P. anomala* NCU003 in the 9% ethanol stress group was significantly higher (*p* < 0.05) for the entire stress period; the highest content of ROS in the 9% ethanol group was determined at 6 h, which was 3.04 times higher than that of the control group.

### 3.7. Effect of Ethanol Stress on the Antioxidant Enzymes Activity of P. anomala NCU003

As shown in Table 2, with the increase in ethanol stress time, the activities of antioxidant enzymes (CAT, SOD, APX and GR) first increased and then declined in *P. anomala* NCU003. Among them, the activity of CAT in the 3% ethanol stress group was significantly higher (*p* < 0.05) than that of the control group at 2–6 h. Compared with the control and 3% ethanol stress groups, the CAT activities of *P. anomala* NCU003 in the 6% and 9% ethanol stress groups were significantly higher (*p* < 0.05) for the entire stress period; the highest activity of CAT was determined in the 9% ethanol stress group at 6 h, which was 4.50 times higher than that of the control group.

The activities of SOD and APX in the 3% ethanol stress group were significantly higher (*p* < 0.05) than in the control group at 4, 6 and 10 h. The activities of SOD and APX in the 6% and 9% ethanol stress groups were significantly higher (*p* < 0.05) than in the control group for the entire stress period. Among them, the highest activity of SOD in the 9% ethanol stress group was determined at 6 h, which was 2.01 times higher than that of the control group, and the highest activity of APX in the 9% ethanol stress group was determined at 8 h, which was 3.52 times higher than that of the control group.

Similarly, the GR activity of *P. anomala* NCU003 in the 6% and 9% ethanol stress groups was significantly higher (*p* < 0.05) than that in the control group for the entire stress period; the highest activity of GR in the 9% ethanol stress group was determined at 6 h, which was 3.48 times higher than that of the control group. Meanwhile, the GR activity of *P. anomala* NCU003 in the 3% ethanol stress group was higher than that of the control group only at 2–6 h, and it was significantly higher (*p* < 0.05) than that of the control group at 6 h. However, the GR activity of *P. anomala* NCU003 in the 3% ethanol stress group was lower than that of the control group after 6 h, and there was no significant difference (*p* > 0.05) between them.

### 3.8. Correlation Analysis of Cell Membrane Integrity and ROS Metabolism

From the correlation analysis results of the control group (Figure 6A), there was no significant correlation relationship between the change in ROS content and the change in antioxidant enzymes activity and MDA content. However, Figure 6B shows that ROS was positively correlated with SOD, CAT, GR and MDA of *P. anomala* NCU003 under 9% ethanol stress; in particular, there was a significant positive correlation between ROS and MDA content. In addition, there was a significant positive or negative correlation between the MDA, electrical conductivity, soluble sugar and soluble protein of *P. anomala* NCU003 in the control and 9% ethanol stress groups.

## 4. Discussion

*P. anomala* as an ester-producing yeast during the brewing process, its normal growth and its esters production ability are the keys to ensure the quality of alcohol products. However, some studies have shown that ethanol stress is an inevitable stress factor for yeast during the brewing process of Chinese *Baijiu*, wine, beer and Japanese sake [14,32]. In our study, it was found that different ethanol stress concentrations could inhibit the growth of *P. anomala* NCU003 both in liquid and solid medium. The higher the ethanol stress concentration, the more serious an inhibitory effect was found (Figure 1). The results of this study were basically consistent with ethanol stress on the growth of *Kluyveromyces marxianus*, *S. cerevisiae* and *Issatchenkia occidentalis*. These studies have also found that ethanol stress could inhibit the growth of these yeasts in liquid and solid medium [17,33].

In addition, our study found that different ethanol stress concentrations could inhibit the production of non-ethyl ester compounds (isoamyl formate, phenylethyl acetate, isopentyl acetate and isobutyl acetate) in *P. anomala* NCU003 (Figure 2). The reduction of these ester compounds’ content may decrease the flavor profiles of honey, flowery and fruity in alcohol products [34]. This was related to the low-growth activity of *P. anomala* NCU003. Similarly, some studies have found that the contents of isoamyl acetate, ethyl hexanoate and higher alcohols decrease in yeast under oxidative and chlorogenic acid stresses [35,36]. However, we also found an interesting phenomenon that the content of ethyl ester compounds (ethyl acetate, ethyl propanoate, ethyl hexanoate, ethyl caprylate and ethyl caprate) increased as the concentration of ethanol increased. This may be due to ethanol and acetyl-CoA being substrates for synthetic ethyl ester compounds. Although ethanol stress inhibited the growth of *P. anomala* NCU003, in turn, it could provide many substrates for the synthesis of ethyl ester compounds [37]. However, the reason for the increase in the ethyl ester compounds content under ethanol stress needs to be further studied.

The integrity of the cell surface morphology is the most basic condition for the normal growth and esters production of yeast. In this study, it was found that the cell surface morphology of *P. anomala* NCU003 changed from plump and smooth to having depressions, or even severe deformation, as the concentrations of ethanol stress increased (Figure 3). This phenomenon was consistent with a study showing that ethanol stress changes the surface morphology of *Issatchenkia orientalis* [25]. This result indicated that a high concentration of ethanol stress could inhibit the growth of *P. anomala* NCU003 by destroying its cell surface morphology. Chen et al. also found that the growth inhibition and cell surface morphology damage of *S. cerevisiae* was observed with the high concentration of ethanol stress [38].

The current study has also found that ethanol stress could inhibit the growth of yeast by destroying the integrity of the cell membrane [39]. The cell membrane is an important part of the yeast response to external environmental factor stress, which plays an important role in maintaining the balance of nutrients (such as sugar and protein) and the intracellular osmotic pressure [40]. Therefore, in order to further explore the mechanism of growth inhibition of *P. anomala* under ethanol stress, the change in the cell membrane integrity of *P. anomala* NCU003 was detected in this study. PI is a DNA binding dye that can be used to detect the integrity of cell membrane because it is not able to pass through the complete cell membrane. However, PI can pass through damaged cell membrane, causing the cell to stain red [41]. At present, PI staining experiments have been widely used to detect the integrity of cell membrane in various cells [42]. Our study found that the number of red cells of *P. anomala* NCU003 gradually increased with increasing ethanol stress concentration (Figure 4). The results indicated that ethanol stress could destroy the integrity of cell membrane, resulting in PI entering the *P. anomala* NCU003 cells through the damaged cell membrane.

The MDA is the main product of plasma membrane lipid peroxidation in the cell membrane. Excessive production of MDA means the damage of cell membrane and can inhibit the normal growth of cells [27]. Our study found that the contents of MDA in *P. anomala* NCU003 under different ethanol stress concentrations were always higher than in the control group, and increased with the increase in the ethanol concentration (Table 1). Another study also found that an increase in plasma membrane lipid peroxidation in *S. cerevisiae* could promote an increase in MDA content in the cell membrane, and damage the integrity of the cell membrane, resulting in inhibiting the growth of *S. cerevisiae* [43]. These results indicated that ethanol stress might destroy the cell membrane integrity of *P. anomala* NCU003 by generating excessive MDA.

In addition, the damage of cell membrane integrity can also cause the flow of intracellular nutrients and electrolytes, resulting in a lack of nutrition and an osmotic pressure imbalance of yeast, which in turn affects the normal growth of yeast. Our study found that the contents of extracellular soluble sugar and soluble protein in *P. anomala* NCU003 under the ethanol stress were significantly increased compared with the control group; the electrical conductivity of *P. anomala* NCU003 also exhibited an increase trend with the increase in ethanol stress concentration (Table 1). These results were consistent with the change in the extracellular leakage substance content of *Millerozyma farinose* under ethanol stress [39]. It was also found that the leakage of intracellular substances was caused by a decline of cell membrane integrity, which was one of the important reasons inhibiting growth of *S. cerevisiae* [19]. Interestingly, we found that the contents of extracellular soluble sugar and soluble protein showed a decline trend with the increase in ethanol stress time. This may be because *P. anomala* NCU003 with normal growth ability continued to absorb and use soluble sugar and soluble protein in the medium, thereby causing this phenomenon [44]. On the other hand, this also may be because the anti-ethanol stress mechanism of *P. anomala* NCU003 was activated under ethanol stress, resulting in it enhancing the repair ability of the cell membrane and thus reducing the contents of extracellular leakage substances. All in all, the current results indicated that ethanol stress could damage the integrity of the cell membrane of *P. anomala* NCU003 by promoting the generation of MDA, resulting in increasing leakage of intracellular soluble sugar, soluble protein and electrolytes, thus inhibiting the growth of *P. anomala* NCU003.

Through further study, we found that the ROS content of *P. anomala* NCU003 was positively correlated with MDA content (Figure 6). At present, some studies have indicated that the accumulation of ROS is one of the significant cellular events in yeast when faced with ethanol stress or other environmental factor stress, and MDA is one of the main products produced by yeast cells due to excessive eruption of ROS [45,46]. ROS is a by-product that is generated due to partial oxygen that cannot be completely reduced during aerobic metabolism process. Excessive ROS can attack biomacromolecules in yeast, such as protein, DNA and lipid, etc., causing oxidation to the cell, the damage of cell membrane integrity and inhibition of the growth of yeast [18]. In this study, we found that the content of ROS increased in *P. anomala* NCU003 at the early stage of ethanol stress, and the higher the ethanol stress concentration, the higher the ROS content (Figure 5). This result was consistent with the study of Jing et al. which also found that ethanol stress could promote the accumulation of ROS content and inhibit the growth of *S. cerevisiae* [15].

However, the ROS content of *P. anomala* NCU003 showed a decrease trend at the late stage of ethanol stress. This may be because *P. anomala* faced the oxidative stress caused by ROS burst, and its own ROS removal system could be activated to eliminate excessive ROS as much as possible. Some studies have found that the improved activities of antioxidant enzymes under the environment factor stress could reduce the accumulation of ROS in yeast, thereby alleviating the effect of environmental factor stress on the growth inhibition of yeast [47,48]. We also found that the activities of SOD, CAT, APX and GR of *P. anomala* NCU003 under ethanol stress were significantly higher than in the control group, and the activities increased with an increase in ethanol concentration (Table 2). Among them, the ASA-GSH cycle is a ROS removal system and is generally present in yeast. GR and APX are key enzymes in ASA-GSH cycles, which both play an important role in removing the excessive ROS in yeast [49]. SOD is also an important antioxidant enzyme that can convert O^2−·^ to lower toxic H_2_O_2_, and CAT can further decompose H_2_O_2_ into H_2_O and O_2_, thereby reducing the damaging effect of ROS on yeast [50]. It was also shown that *S. cerevisiae* could alleviate the damage to cell membrane by reducing the accumulation of ROS content and increasing the activities of CAT and SOD under high osmotic pressure stress [51]. Of note, another study has also indicated that the activities of SOD, CAT and GR of *S. cerevisiae* decreased under other stress conditions [52]. These results indicated that the damage mechanism in different yeast strains may be different when facing the stress factor environment.

At present, the excess ROS produced under ethanol stress plays an important role in inhibiting the growth metabolism and non-ethyl ester production ability of *P. anomala*. It not only increased the permeability of cell membrane, but also activated the activity of antioxidant enzymes in response to the damage caused by ROS (Figure 7). However, the current results indicated that ethanol stress activated the activities of these antioxidant enzymes, but they were still insufficient to completely reduce the damage to cell membrane integrity caused by the ROS burst. Therefore, more information on changes in antioxidant enzymes activity and ROS content is needed in order to better understand the relation between the growth of *P. anomala* and ethanol stress.

## 5. Conclusions

As we know, this is the first study about ethanol stress effects on the growth and esters production ability of *P. anomala*, revealing the regulation mechanisms under ethanol stress. This study found that ethanol stress had the capability to inhibit the growth and the non-ethyl ester compounds production of *P. anomala* NCU003; the inhibitory effects were more obvious with the increase in ethanol stress concentration. Moreover, the influence mechanism may be due to ethanol stress causing ROS burst in *P. anomala* NCU003, thereby promoting the accumulation of MDA content in the cell membrane, destroying the integrity of the cell membrane and leading to the leakage of intracellular nutrients and electrolytes. Finally, the growth and non-ethyl ester production ability of *P. anomala* NCU003 was inhibited. All in all, this work can provide an important theoretical reference for further research on the mechanism affecting the growth and esters production ability of *P. anomala* under ethanol stress. In addition, we also proved that ethanol stress has the potential capability to change the content of ester compounds in *P. anomala*. In the future, this will play an important role in guiding the application of *P. anomala* in the alcohol products brewing industry.

## Figures and Tables

**Figure 1 foods-11-03744-f001:**
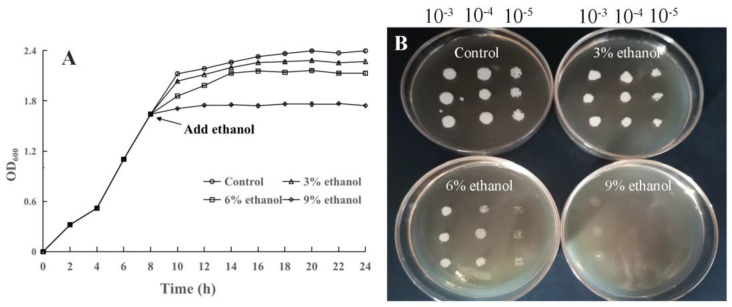
Effects of ethanol stress on the growth of *P. anomala* NCU003 in liquid medium (**A**) and solid medium (**B**).

**Figure 2 foods-11-03744-f002:**
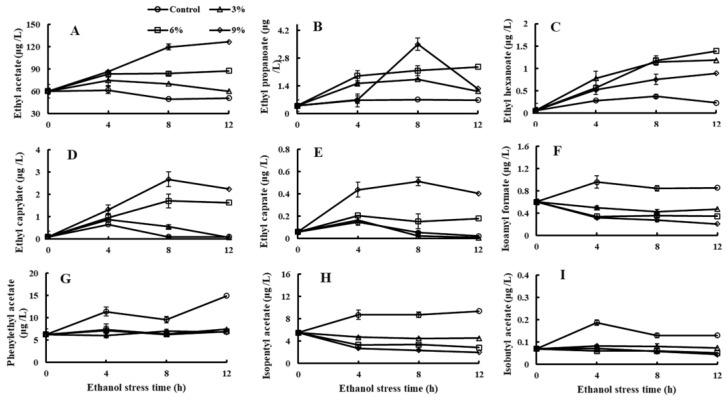
Effects of ethanol stress on the content of ethyl acetate (**A**), ethyl propanoate (**B**), ethyl hexanoate (**C**), ethyl caprylate (**D**), ethyl caprate (**E**), isoamyl formate (**F**), phenylethyl acetate (**G**), isopentyl acetate (**H**) and isobutyl acetate (**I**) of *P. anomala* NCU003.

**Figure 3 foods-11-03744-f003:**
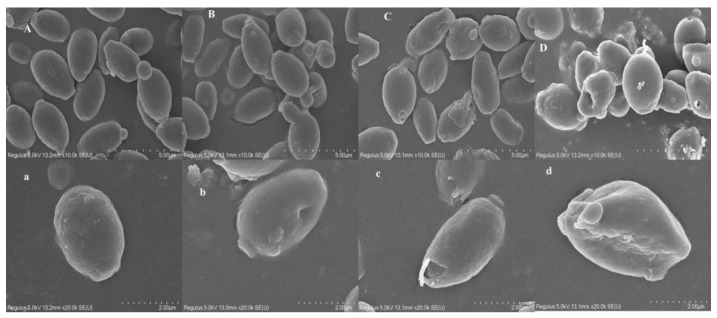
SEM images of *P. anomala* NCU003 under different ethanol stress concentrations ((**A**), (**a**): control; (**B**), (**b**): 3% ethanol; (**C**), (**c**): 6% ethanol; (**D**), (**d**): 9% ethanol; (**A**–**D**): 10×; (**a**–**d**): 20×).

**Figure 4 foods-11-03744-f004:**
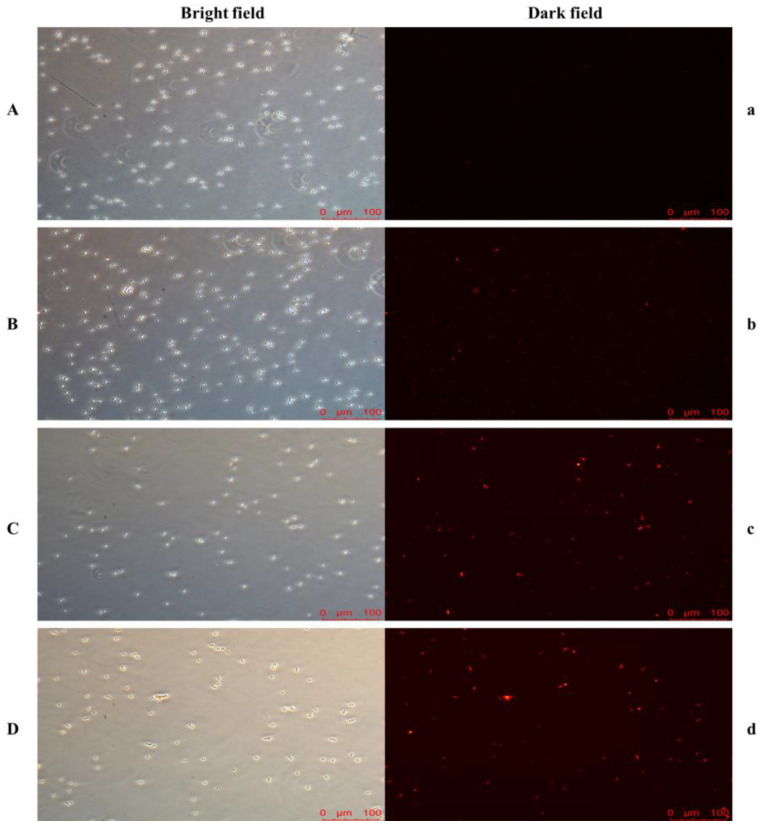
The PI staining results of *P. anomala* NCU003 under different ethanol stress concentrations ((**A**), (**a**): control; (**B**), (**b**): 3% ethanol; (**C**), (**c**): 6% ethanol; (**D**), (**d**): 9% ethanol).

**Figure 5 foods-11-03744-f005:**
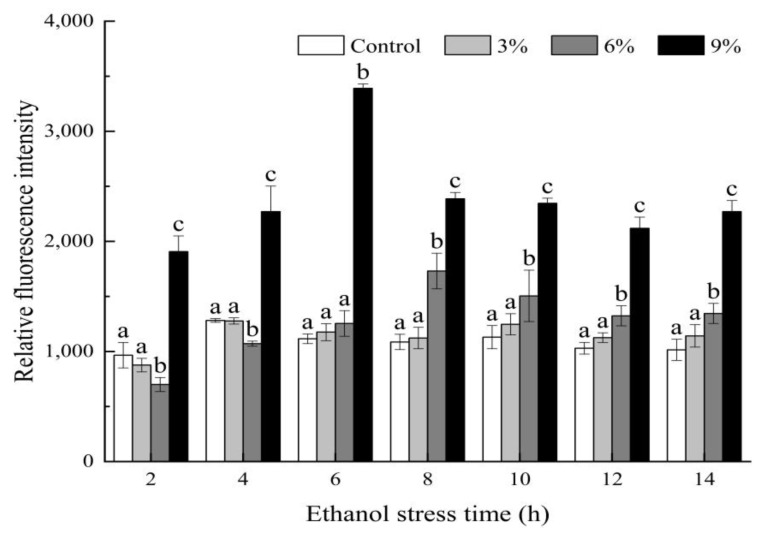
Effects of ethanol stress on ROS content of *P. anomala* NCU003 (different alphabet represents a significant difference (*p* < 0.05) at the same time).

**Figure 6 foods-11-03744-f006:**
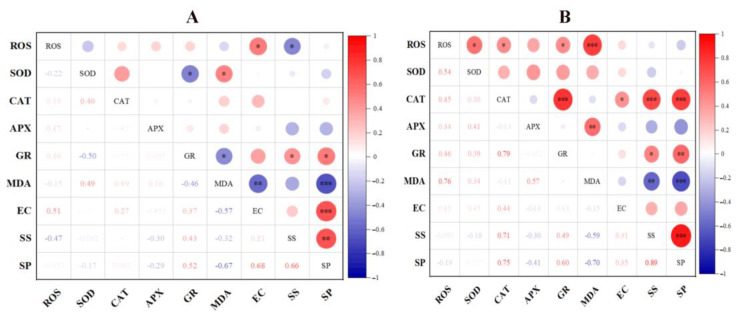
Correlation analysis of cell membrane integrity and ROS-metabolism-related indicators of the control (**A**) and 9% ethanol stress (**B**) groups in *P. anomala* NCU003 (EC: electrical conductivity; SS: soluble sugar; SP: soluble protein; *: *p* < 0.05; **: *p* < 0.05; ***: *p* < 0.001).

**Figure 7 foods-11-03744-f007:**
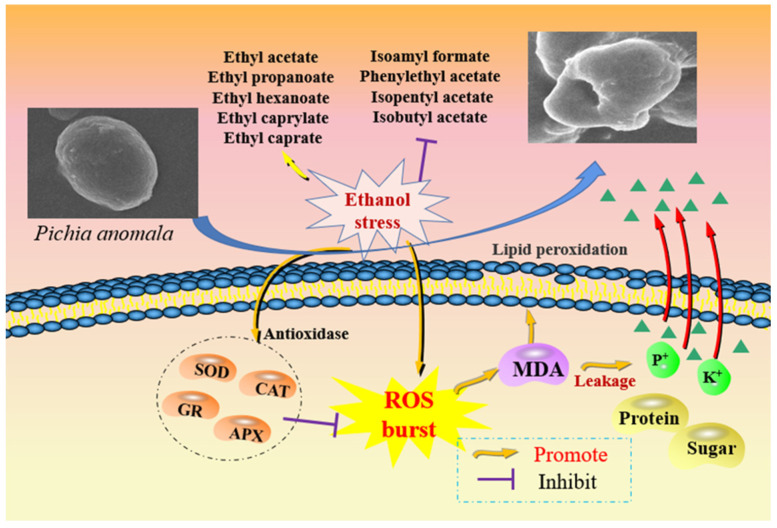
Schematic diagram of the influence mechanism of ethanol stress on *P. anomala*.

**Table 1 foods-11-03744-t001:** Effects of different concentrations of ethanol stress on the MDA and extracellular leakage substances content of *P. anomala* NCU003.

	Samples	Stress Time (h)							
		0	2	4	6	8	10	12	14
MDA content (μmol/g)	Control	2.19 ± 0.59 ^a^	2.97 ± 1.05 ^a^	3.84 ± 1.66 ^a^	5.03 ± 0.47 ^a^	5.17 ± 0.69 ^a^	4.54 ± 2.05 ^a^	5.59 ± 0.40 ^a^	6.73 ± 1.68 ^a^
	3%	2.19 ± 0.59 ^a^	4.72 ± 0.96 ^b^	5.63 ± 0.28 ^b^	5.32 ± 0.91 ^a^	5.76 ± 0.98 ^a^	5.65 ± 0.48 ^a^	7.56 ± 1.26 ^b^	8.15 ± 0.53 ^a^
	6%	2.19 ± 0.59 ^a^	5.02 ± 1.14 ^b^	9.94 ± 2.10 ^c^	10.41 ± 3.03 ^b^	9.43 ± 0.90 ^b^	7.93 ± 1.94 ^b^	11.34 ± 2.66 ^c^	12.84 ± 1.54 ^b^
	9%	2.19 ± 0.59 ^a^	21.71 ± 3.07 ^c^	24.34 ± 4.18 ^d^	29.16 ± 1.58 ^c^	28.19 ± 2.64 ^c^	26.04 ± 2.11 ^c^	26.16 ± 1.99 ^d^	26.81 ± 2.12 ^c^
Electrical conductivity (mS/cm)	Control	25.63 ± 0.86 ^a^	26.75 ± 0.56 ^a^	28.23 ± 0.70 ^a^	28.09 ± 0.47 ^a^	24.78 ± 0.18 ^a^	24.58 ± 0.23 ^a^	24.24 ± 0.27 ^a^	23.37 ± 0.89 ^a^
	3%	25.63 ± 0.86 ^a^	28.09 ± 0.24 ^a^	32.66 ± 1.50 ^a^	29.72 ± 0.44 ^a^	28.07 ± 0.65 ^b^	25.92 ± 0.34 ^a^	26.25 ± 1.35 ^a^	27.37 ± 0.74 ^b^
	6%	25.63 ± 0.86 ^a^	35.58 ± 1.01 ^b^	36.08 ± 0.44 ^a^	33.77 ± 0.43 ^b^	34.78 ± 0.63 ^c^	32.49 ± 0.62 ^b^	35.25 ± 1.55 ^b^	36.40 ± 1.07 ^c^
	9%	25.63 ± 0.86 ^a^	85.95 ± 3.69 ^c^	83.85 ± 8.29 ^b^	85.29 ± 1.88 ^c^	81.66 ± 2.18 ^d^	79.87 ± 1.17 ^c^	79.12 ± 2.28 ^c^	83.92 ± 3.87 ^d^
Soluble protein content (μg/g)	Control	20.14 ± 0.15 ^a^	10.10 ± 0.48 ^a^	8.57 ± 0.08 ^a^	8.40 ± 0.07 ^a^	7.34 ± 0.08 ^a^	7.70 ± 0.13 ^a^	6.96 ± 0.08 ^a^	7.38 ± 0.60 ^a^
	3%	20.14 ± 0.15 ^a^	11.06 ± 0.21 ^b^	10.28 ± 0.09 ^b^	9.54 ± 0.16 ^b^	8.54 ± 0.27 ^b^	8.01 ± 0.15 ^a^	8.09 ± 0.21 ^b^	8.16 ± 0.27 ^b^
	6%	20.14 ± 0.15 ^a^	14.05 ± 0.61 ^c^	13.44 ± 0.15 ^c^	12.06 ± 0.34 ^c^	11.35 ± 0.19 ^c^	10.76 ± 0.05 ^b^	11.20 ± 0.09 ^c^	11.28 ± 0.26 ^c^
	9%	20.14 ± 0.15 ^a^	35.61 ± 0.24 ^d^	32.63 ± 0.71 d	29.00 ± 0.23 ^d^	27.62 ± 0.78 ^d^	26.42 ± 0.16 ^c^	25.68 ± 0.16 ^d^	24.75 ± 0.62 ^d^
Soluble sugar content (mg/g)	Control	25.16 ± 0.93 ^a^	3.68 ± 1.55 ^a^	0.92 ± 0.17 ^a^	1.19 ± 0.45 ^a^	1.31 ± 0.19 ^a^	1.18 ± 0.60 ^a^	1.13 ± 0.12 ^a^	0.88 ± 0.10 ^a^
	3%	25.16 ± 0.93 ^a^	6.46 ± 0.09 ^b^	2.04 ± 0.53 ^a^	2.79 ± 0.86 ^a^	1.97 ± 0.24 ^a^	1.30 ± 0.12 ^a^	2.55 ± 0.21 ^a^	1.46 ± 0.21 ^a^
	6%	25.16 ± 0.93 ^a^	11.00 ± 1.48 ^c^	10.21 ± 1.32 ^b^	4.06 ± 0.62 ^b^	2.57 ± 0.27 ^b^	3.47 ± 1.65 ^b^	4.52 ± 0.97 ^b^	1.97 ± 0.39 ^a^
	9%	25.16 ± 0.93 ^a^	35.42 ± 2.87 ^d^	22.98 ± 1.04 ^c^	23.90 ± 1.13 ^c^	20.08 ± 2.52 ^c^	17.78 ± 2.90 ^c^	17.96 ± 1.21 ^c^	10.15 ± 1.62 ^b^

Note: different superscripts in the same column indicate significant difference (*p* < 0.05) among control and ethanol stress groups.

**Table 2 foods-11-03744-t002:** Effects of different concentration of ethanol stress on the antioxidant enzymes activity of *P. anomala* NCU003.

	Samples	Stress Time (h)							
		0	2	4	6	8	10	12	14
CAT (U/g)	Control	5.62 ± 0.52 ^a^	5.55 ± 0.08 ^a^	5.85 ± 0.40 ^a^	5.94 ± 0.87 ^a^	5.89 ± 0.17 ^a^	4.63 ± 0.23 ^a^	5.24 ± 0.13 ^a^	5.42 ± 1.11 ^a^
	3%	5.62 ± 0.52 ^a^	6.66 ± 0.29 ^b^	7.27 ± 0.86 ^b^	7.50 ± 0.48 ^b^	6.33 ± 0.51 ^a^	5.48 ± 0.91 ^a^	5.68 ± 0.19 ^a^	6.22 ± 1.00 ^a^
	6%	5.62 ± 0.52 ^a^	9.30 ± 0.46 ^c^	8.88 ± 0.29 ^c^	10.02 ± 0.62 ^c^	9.64 ± 0.20 ^b^	9.81 ± 0.47 ^b^	7.93 ± 0.33 ^b^	7.76 ± 0.15 ^b^
	9%	5.62 ± 0.52 ^a^	25.14 ± 0.16 ^d^	25.02 ± 0.59 ^d^	26.71 ± 0.89 ^d^	22.63 ± 0.61 ^c^	20.40 ± 0.73 ^c^	19.37 ± 1.52 ^c^	19.14 ± 1.07 ^c^
SOD (U/g)	Control	12.64 ± 0.84 ^a^	13.24 ± 0.51 ^a^	12.50 ± 0.09 ^a^	16.58 ± 0.07 ^a^	13.46 ± 0.09 ^a^	12.32 ± 0.16 ^a^	14.96 ± 0.10 ^a^	14.26 ± 0.73 ^a^
	3%	12.64 ± 0.84 ^a^	14.66 ± 0.23 ^a^	17.37 ± 0.09 ^b^	18.62 ± 0.16 ^b^	13.54 ± 0.29 ^a^	14.76 ± 0.23 ^ab^	15.47 ± 0.29 ^a^	16.56 ± 0.35 ^ab^
	6%	12.64 ± 0.84 ^a^	16.96 ± 0.68 ^b^	18.82 ± 0.19 ^b^	22.82 ± 0.35 ^c^	16.13 ± 0.23 ^b^	18.82 ± 0.06 ^b^	19.78 ± 0.10 ^b^	19.69 ± 0.40 ^b^
	9%	12.64 ± 0.84 ^a^	25.05 ± 0.14 ^c^	29.68 ± 0.64 ^c^	33.26 ± 0.20 ^d^	26.90 ± 0.70 ^c^	27.22 ± 0.16 ^c^	24.58 ± 0.16 ^c^	28.58 ± 0.54 ^c^
APX (U/g)	Control	0.58 ± 0.14 ^a^	0.54 ± 0.07 ^a^	0.61 ± 0.06 ^a^	0.62 ± 0.15 ^a^	0.69 ± 0.06 ^a^	0.61 ± 0.11 ^a^	0.62 ± 0.09 ^a^	0.61 ± 0.09 ^a^
	3%	0.58 ± 0.14 ^a^	0.63 ± 0.10 ^a^	0.85 ± 0.25 ^b^	0.83 ± 0.26 ^b^	0.75 ± 0.08 ^a^	0.79 ± 0.07 ^b^	0.63 ± 0.18 ^a^	0.60 ± 0.06 ^a^
	6%	0.58 ± 0.14 ^a^	0.67 ± 0.11 ^a^	0.96 ± 0.10 ^b^	1.13 ± 0.23 ^c^	1.35 ± 0.09 ^b^	1.19 ± 0.09 ^c^	0.83 ± 0.24 ^b^	0.76 ± 0.22 ^b^
	9%	0.58 ± 0.14 ^a^	1.73 ± 0.35 ^b^	1.85 ± 0.03 ^c^	2.23 ± 0.28 ^d^	2.43 ± 0.22 ^c^	2.11 ± 0.27 ^d^	2.06 ± 0.30 ^c^	1.99 ± 0.26 ^c^
GR (U/g)	Control	0.37 ± 0.06 ^a^	0.40 ± 0.07 ^a^	0.39 ± 0.06 ^a^	0.29 ± 0.12 ^a^	0.37 ± 0.05 ^a^	0.31 ± 0.03 ^a^	0.27 ± 0.03 ^a^	0.33 ± 0.07 ^a^
	3%	0.37 ± 0.06 ^a^	0.47 ± 0.09 ^ab^	0.49 ± 0.09 ^ab^	0.45 ± 0.06 ^b^	0.35 ± 0.03 ^a^	0.26 ± 0.02 ^a^	0.25 ± 0.02 ^a^	0.32 ± 0.03 ^a^
	6%	0.37 ± 0.06 ^a^	0.61 ± 0.09 ^b^	0.63 ± 0.04 ^b^	0.48 ± 0.05 ^bc^	0.44 ± 0.11 ^b^	0.39 ± 0.04 ^b^	0.35 ± 0.04 ^b^	0.37 ± 0.09 ^a^
	9%	0.37 ± 0.06 ^a^	0.86 ± 0.11 ^c^	0.93 ± 0.19 ^c^	1.01 ± 0.17 ^c^	0.77 ± 0.18 ^c^	0.67 ± 0.09 ^c^	0.60 ± 0.17 ^c^	0.63 ± 0.08 ^b^

Note: different superscripts in the same column indicate significant difference (*p* < 0.05) among control and ethanol stress groups.

## Data Availability

The data presented in this study are available on request from the corresponding author.

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
