# Peer review of "Effects on Cell Membrane Integrity of Pichia anomala by the Accumulating Excessive Reactive Oxygen Species under Ethanol Stress"

_foods, 2022, doi:10.3390/foods11223744_

Round 1

Reviewer 1 Report

Contains information of interest to readers. However, some areas must be improved especially the following areas:

Abstract needs minor editing - technical editing. Conclusion is not clear - needs editing.

Materials and Methods; Results. These two sections are challenged by language. Authors need to seek assistance to edit the paper. Technically, it is not preferred to start a statement by a number, e..g. 10 mL, etc. but you can consider writing as Ten (10) mL ................ There are numerous half statements in this section that need to be corrected. Centrifugation is written in centrifugal force (g) and not rpm. All materials and equipment must state the suppliers/origin, etc. Refer graphs to: y and x - axis where necessary.  Nomenclature of microorganisms should follow the international system. Also, do not start written statements with, 'And ..........'.

Tables 1 and 2 with stats: revise the note as acceptable to statistical terms: different superscripts in the same column indicate sig. differences at xxxxx

The term, at the same time - needs to be corrected.

Abstract needs minor editing - technical editing.

Author Response

Dear reviewer

Thank you for your consideration of our manuscript (2004010). We have revised the manuscript according to your comments step by step. We hope the manuscript is now suitable for Foods.

Point 1: Abstract needs minor editing - technical editing. Conclusion is not clear - needs editing.

Response 1: Thank you for your suggestion. We have revised the Abstract and Conclusion section according to your suggestion. (Line 14-16; 465-474)

Point 2: Materials and Methods; Results. These two sections are challenged by language. Authors need to seek assistance to edit the paper. Technically, it is not preferred to start a statement by a number, e..g. 10 mL, etc. but you can consider writing as Ten (10) mL ................ There are numerous half statements in this section that need to be corrected. Centrifugation is written in centrifugal force (g) and not rpm. All materials and equipment must state the suppliers/origin, etc. Refer graphs to: y and x - axis where necessary.  Nomenclature of microorganisms should follow the international system. Also, do not start written statements with, 'And ..........'.

Response 2: Thank you for your suggestion. We have revised the Materials and Methods and Results section according to your suggestion.

Point 3: Tables 1 and 2 with stats: revise the note as acceptable to statistical terms: different superscripts in the same column indicate sig. differences at xxxxx

Response 3: Thank you for your suggestion. The sentence has revised as “different superscripts in the same column indicate significant difference (p<0.05) among control and ethanol stress groups”. (Line 241-243)

Point 4: The term, at the same time - needs to be corrected.

Response 4: Thank you for your suggestion. The sentence has revised as “different superscripts in the same column indicate significant difference (p<0.05) among control and ethanol stress groups”. (Line 241-243)

Point 5: Abstract needs minor editing - technical editing.

Response 5: Thank you for your suggestion. We have revised the Abstract section according to your suggestion. (Line 14-16)

Reviewer 2 Report

The manuscript “Effect of cell membrane integrity of Pichia anomala by the accumulating excessive reactive oxygen species under ethanol stress” is well written. Following modifications:

1.       Author have discussed the effect of 7 different parameters (section 3.1 to 3.7). Can author explain which parameter is more important and have more effect.

2.       The objective of the study should be presented in a detail and more clear way.

3.       Section 2.1 Line 75-77. The method ref. [11] should be written/explained for better understanding.

4.       A schematic diagram of experimental setup must be provided.

5.       A separate abbreviation list should be provided.

6.       Author should provide better quality images for Figures 1(A), 2, 6.

7.       Figure 3and 4. Scale is not clear.

8.       Figure 7 should be placed and discussed at the end of section 4.

Author Response

Dear reviewer

Thank you for your consideration of our manuscript (2004010). We have revised the manuscript according to your comments step by step. We hope the manuscript is now suitable for Foods.

The manuscript “Effect of cell membrane integrity of Pichia anomala by the accumulating excessive reactive oxygen species under ethanol stress” is well written. Following modifications:

Point 1: Author have discussed the effect of 7 different parameters (section 3.1 to 3.7). Can

author explain which parameter is more important and have more effect.

Response 1: Thank you for your suggestion. We think ROS is the most important indicator in

this study and we have revised the discussion section according to your suggestion. (Line 445-448)

Point 2: The objective of the study should be presented in a detail and more clear way.

Response 2: Thank you for your suggestion. We have revised the objective of the study

according to your suggestion. (Line 74-80).

Point 3: Section 2.1 Line 75-77. The method ref. [11] should be written/explained for better

understanding.

Response 3: Thank you for your suggestion. We have revised this section according

to your suggestion. (Line 84).

Point 4: A schematic diagram of experimental setup must be provided.

Response 4: Thank you for your suggestion. We have added the experimental schematic

diagram (Figure 7) in the manuscript.

Point 5: A separate abbreviation list should be provided.

Response 5: Thank you for your suggestion. We have added abbreviations list in the

manuscript. (Line 489-492)

Point 6: Author should provide better quality images for Figures 1(A), 2, 6.

Response 6: Thank you for your suggestion. We have modified Figure 1, Figure 2 and Figure

6 according to your suggestion.

Point 7: Figure 3and 4. Scale is not clear.

Response 7: Thank you for your suggestion. We have modified Figure 3 and Figure 4

according to your suggestion.

Point 8: Figure 7 should be placed and discussed at the end of section 4.

Response 8: Thank you for your suggestion. Figure 7 has been placed in the discussion

section.
